# ProSPective evaluation of the dIagnostic accuracy of siNe spiN non-contrast flatdEtectoR CT (FDCT) for the detection of intracranial hemorrhage in stroke patients - Protocol of a non-inferiority comparison to multi detector CT

Marios Psychogios[1◉*], Alex Brehm[1◉], Nitin Goyal[2], Grégoire Boulouis[3], Jan-Karl Burkhardt[4], Shakeel A. Chowdhry[5], Donald Frei[6], Jan Gralla[7], Johannes Kaesmacher[3,7], Ryan T. Kellogg[8], Christopher P. Kellner[9], Demetrius K. Lopes[10], Eytan Raz[11], Daniel Strbian[12], Laura Mannismäki[12], Alejandro Tomasello[13], Ioannis Tsogkas[1], Alexander von Hessling[14], Grzegorz M. Karwacki[14], Raphael Guzman[15], Nikki Rommers[16], David S. Liebeskind[17], Adam S. Arthur[2], on behalf of the SPINNERS investigators[¶]

1 Department of Diagnostic and Interventional Neuroradiology, University Hospital Basel, Basel, Switzerland, 2 Department of Neurosurgery, Semmes Murphey Clinic and University of Tennessee Health Sciences Center, Memphis, Tennessee, United States of America, 3 Service de Neuroradiologie, CHRU Tours, Tours, France, 4 Department of Cerebrovascular Surgery, Hospital of the University of Pennsylvania, Philadelphia, Pennsylvania, United States of America, 5 Department of Neurosurgery, Endeavor Health, Chicago, Illinois, United States of America, 6 RIA Neurovascular, Englewood, Colorado, United States of America, 7 Department of Neuroradiology, Inselspital Bern, Bern, Switzerland, 8 Department of Neurosurgery, University of Virginia, Charlottesville, Virginia, United States of America, 9 Department of Neurosurgery, Mt Sinai Hospital Systems, New York, New York, United States of America, 10 Department of Neurosurgery, Advocate Medical Group, Park Ridge, Illinois, United States of America, 11 Department of Neuroradiology, NYU Langone Health, New York, New York, United States of America, 12 Department of Neurology, Helsinki University Hospital and University of Helsinki, Helsinki, Finland, 13 Department of Neuroradiology, Hospital Vall d'Hebron, Barcelona, Spain, 14 Section of Neuroradiology, Radiology and Nuclear Medicine, Cantonal Hospital of Lucerne, Lucerne, Switzerland, 15 Department of Neurosurgery, University Hospital Basel, Basel, Switzerland, 16 Department of Clinical Research, University Hospital Basel, Basel, Switzerland, 17 Department of Neurology, UC Los Angeles, Los Angeles, California, United States of America

◉ These authors contributed equally to this work.
¶ Collaborators are listed in the Acknowledgments and in the supplementary Appendix.
* Marios.psychogios@usb.ch

## Abstract

### Rationale

Whether syngo DynaCT Sine Spin non-contrast flat detector CT (FDCT) imaging is sufficient to rule out intracranial hemorrhage in suspected acute stroke patients is unknown.

### Aim

To determine if syngo DynaCT Sine Spin non-contrast FDCT imaging is non-inferior to conventional multidetector CT (MDCT) imaging for the detection and exclusion of intracranial hemorrhages in suspected acute stroke patients.

**Data availability statement:** Deidentified research data may be made publicly available when the study is completed and published.

**Funding:** A grant from Siemens Healthineers AG was given to the University Hospital Basel (Sponsor Investigator Marios Psychogios) for conducting the trial as an investigator-initiated trial. Durin the planning phase of the trial experts nominated by and employees of Siemens Healthineers AG were consulted, but the Global PI (Marios Psychogios) and the US PI (Adam Arthur) had the final decision to all suggestions. Siemens Healthineers AG had no role in the collection, monitoring, analysis, and interpretation of the data or scientific interpretation and publication of the results.

**Competing interests:** Grant from Siemens Healthineers AG for conducting the trial as an investigator initiated trial. This does not alter our adherence to PLOS ONE policies on sharing data and materials.

## Sample size

To enroll 252 participants in three buckets (126 ischemic stroke patients, 126 hemorrhagic stroke patients (including 14 patients with an isolated infratentorial hemorrhage).

## Methods and design

A multicenter, international, prospective, cross-sectional, endpoint assessor blinded, non-inferiority trial.

## Outcomes

The primary outcome is the occurrence of an intracranial hemorrhage (yes versus no). This will be used to calculate the sensitivity and specificity of FDCT imaging for the detection of intracranial hemorrhages. All FDCT images will be rated by six independent raters in a blinded imaging core-lab. The rating of the MDCT images will be deemed as ground-truth. FDCT imaging will be deemed non-inferior if the lower bound of the 95%-Confidence Interval of the sensitivity and specificity is above 95%.

## Discussion

This trial will inform physicians whether syngo DynaCT Sine Spin non-contrast FDCT imaging can reliably exclude intracranial hemorrhages in patients with suspected acute stroke.

## Trial registration

ClinicalTrials.gov NCT05458908

---

## Introduction

Endovascular treatment (EVT) has dramatically improved the outcome of acute ischemic stroke patients with a large vessel occlusion (LVO) [1]. However, as was shown by a post-hoc meta-analysis of the pivotal LVO trials, clinical outcome is highly associated with time from hospital admission to reperfusion [2]. Implementing a direct-to-angiography workflow may substantially reduce intrahospital delays. In this approach, both imaging with non-contrast flat detector CT (FDCT) to exclude intracranial hemorrhage and subsequent EVT is performed in the angiography suite. A direct to angiography workflow can dramatically reduce intra-hospital time delays (median reductions of more than 30 minutes) and is associated with improved patient outcomes [3–5]. A major obstacle to the widespread implementation of a direct to angiography approach is the uncertainty whether non contrast FDCT can reliably distinguish ischemic from hemorrhagic strokes. To date, evidence has been confined to retrospective analyses and to peri-interventional hemorrhages, which have imaging characteristics that differ markedly from those of spontaneously occurring intracranial hemorrhages.

The "ProSPective evaluation of the dIagnostic accuracy of siNe spiN non-contrast flatdEtectoR CT (FDCT) for the detection of intracranial hemorrhage in Stroke patients" (SPINNERS) study will be the first study to address this question prospectively in the relevant patient population, which is patients presenting with a prehospital stroke code activation.

## Methods

### Study design

The SPINNERS trial is an investigator-initiated, multicenter, international, prospective, cross-sectional, assessor blinded, non-inferiority study (current protocol version 2.1, dated 29.04.2024). The trial was registered prior to the inclusion of the first patient on ClinicalTrials.gov (identifier NCT05458908). The investigational device under study is non-contrast syngo DynaCT Sine Spin FDCT imaging (hereinafter referred to as Sine Spin FDCT imaging) and the control intervention is non-contrast multidetector CT (MDCT) imaging. The hypothesis is that Sine Spin FDCT imaging is non-inferior for the detection of intra-cranial hemorrhages compared to non-contrast MDCT imaging. Participants are enrolled after MDCT imaging and best med-ical treatment if a second scan with Sine Spin FDCT is feasible within four hours of the MDCT, and no invasive intervention is planned in between. Enrollment prior to the non-contrast MDCT, and best medical treatment would lead to unacceptable time delays in this patient cohort. Based on clinical symptoms and imaging findings of the non-contrast MDCT, patients can be enrolled in one of three buckets: (1) acute ischemic stroke, (2) an intracranial hemorrhage which has supratentorial portions and (3) an isolated infratentorial or basal intracranial hemorrhage (including perimesencephalic subarachnoid hemorrhage [SAH]). A total of 252 participants will be enrolled, distributed over the buckets in the following proportions:

- Bucket 1 (acute ischemic stroke patients): 126 participants

- Bucket 2 (patients with an intracranial hemorrhage with supratentorial portions): 112 participants

- Bucket 3 (isolated infratentorial or basal intracranial hemorrhage): 14 participants

The schedule of assessments is shown in Fig 1 and participant flow is depicted in Fig 2. In brief, participants are screened after emergency imaging with multidetector CT and start of best medical treatment. All treatment decisions should be based on the initial multidetector CT. If they fulfill all inclusion and none of the exclusion criteria, participants are enrolled. After enrollment Sine Spin FDCT imaging is done in the angiography suite. Imaging must be done prior to any invasive intervention and within 4 hours of the multidetector CT scan. The Sine Spin FDCT imaging is the only study specific intervention and participants will be followed up for a safety period of 24 hours to identify device related adverse events. Written or oral informed consent was obtained from all participants or their legally authorized representatives either before the study or post hoc (emergency consent procedures), according to country-specific requirements. In case of verbal consent this was documented on a specific form. The trial is conducted in accordance with the ISO 14155 and the Declaration of Helsinki. The study was approved by the Ethics Commission Northwest- and Zentralswitzerland, Ethics Commission of Berne (Switzerland), HUS Regional Medical Research Ethics Committee (Finland), Ethics of research with medicines and research projects committee of the hospital Universitario Vall d'hebron (Spain), Comité de protection des personnes Sud-Est III (France), Central IRB WCG (USA), Institutional Review Board (IRB) of Northshore Research Insti-tute (USA), IRB of the Mount Sinai School of Medicine (USA) and the IRB of NYU Langone Health. The FDCT and MDCT images are collected centrally and will be rated in a blinded independent core-lab. The trial is being conducted in over 10 comprehensive stroke centers in Switzerland, Finland, the United States of America, France and Spain (see list of sites in the appendix). The first patient was enrolled on the 25th of October 2022.

### Population

The trial population consists of patients who present with a severe suspected stroke (defined as a National Institutes of Health Stroke Scale (NIHSS) score of 7 or higher or symptoms suggestive of hemorrhagic stroke) within 24 hours of last

| Visits | | 0 | 1 |
|---|---|---|---|
| | | **Screening and enrollment** | **Intervention** |
| **Assessment** | **Time Window** | Admission | 0 – 4 hours after Non-contrast MDCT scan |
| | **Method** | Clinical Visit | Intervention |
| Informed consent | | X[1] | |
| Patient demographics (i.e. age, gender, pre-existing conditions) | | X | |
| National Institute of Health Stroke Scale | | X | |
| Pregnancy test (only in women of childbearing potential) | | X | |
| Patient logistics (i.e. time from onset to admission / imaging) | | X | |
| Medical history (including medication) | | X | |
| Non-contrast MDCT head scan[2] | | X | |
| Enrollment | | X | |
| Non-contrast syngo DynaCT Sine Spin FDCT head scan | | | X |
| AEs | | Only if procedure related up to 24 hours after procedure | |
| SAEs | | Only if procedure related up to 24 hours after procedure | |

1. Post-hoc consent if patient was not able to give consent at trial inclusion (according to national and applicable law) or FDCT done as per standard of care
2. For clarification the MDCT scan could have been done in an external hospital (in case of transfer patients) if the images of the MDCT are available at the PACS of the enrolling hospital and the quality was deemed to be sufficient by the enrolling physician and all other in- and exclusion criteria are met

AEs Adverse Events, FDCT Flat-Detector CT, MDCT Multi-Detector CT, SAEs Serious Adverse Events

**Fig 1. Schedule of assessments.**

seen well to the enrolling hospital. This includes patients with ischemic and hemorrhagic strokes. The diagnosis of an ischemic or hemorrhagic stroke is made in the acute setting, primarily in the emergency room, prior to patient enrollment. At this stage, clinical evaluation—such as the assessment of acute NIHSS or the Glasgow Coma Scale (GCS) — combined with imaging findings from the MDCT forms the basis for diagnosis. The inclusion and exclusion criteria are listed in Table 1.

## Enrollment and blinding

Participants must be enrolled within 24 hours of last seen well. Participants are enrolled after MDCT imaging if they fulfill all inclusion and none of the exclusion criteria. Enrollment and registration into a web-based data management system are completed prior to the FDCT. For each participant withdrawing consent before the final outcome assessment, an additional participant is included. The primary outcome (occurrence of intracranial hemorrhage) will be assessed by independent and blinded readers in a core lab (provided by the Neurovascular Imaging Research Core, University of California, Los Angeles, USA under the direction of Prof. David Liebeskind). The core lab director is responsible for ensuring that readers are adequately experienced in the interpretation of non-contrast MDCT and FDCT scans of the head.

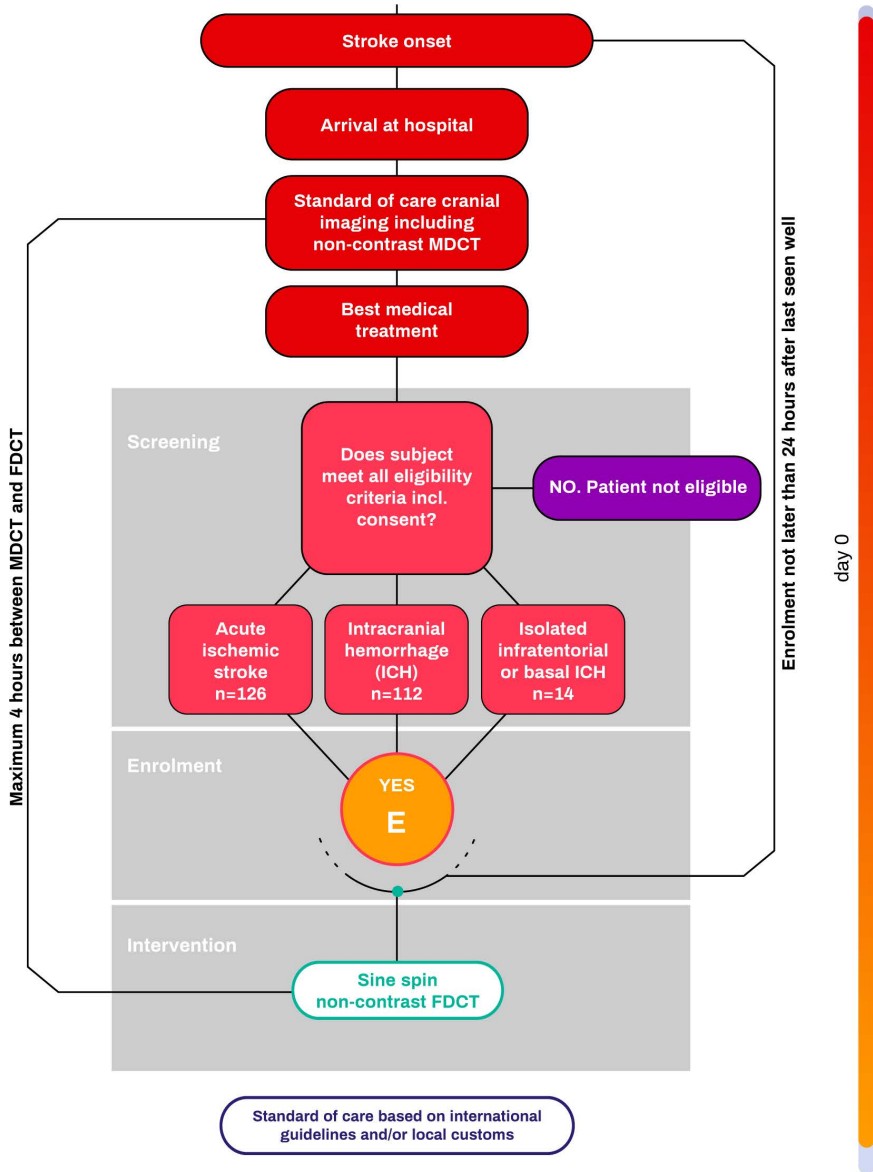

**Fig 2. Patient flow chart.**

## Intervention

The experimental intervention is performed on an ARTIS icono (Siemens Healthineers AG, Forchheim, Germany) biplane angiographic system with non-contrast *syngo* DynaCT Sine Spin scan of the head. *syngo* DynaCT Sine Spin is a standard imaging protocol for visualization of the brain parenchyma and skull. The scan duration is approximately seven seconds, and the effective dose is up to 2.5 mSv [6]. The control intervention is a non-contrast MDCT.

## Assessment of primary outcome

The primary outcome is the presence of an intracranial hemorrhage (yes versus no), as assessed by an independent, blinded core lab (i.e., all readers are blinded to all clinical information). The ground truth (i.e., if an intracranial hemorrhage

**Table 1. In- and exclusion criteria.**

| Inclusion criteria: |
| --- |
| 1. Patients with symptoms suggestive of ischemic stroke (NIHSS ≥ 7) or suggestive of hemorrhagic stroke with a cranial non-contrast MDCT and a feasible non-contrast syngo DynaCT Sine Spin within 4 hours |
| 2. Patient presenting within 24 hours of last-seen well |
| 3. Patients presenting directly to the treating hospital (i.e., mothership patients) OR transfer patients with the indication for repeated imaging according to the standard operation procedures of the treating hospital* |
| 4. Age ≥ 18 years |
| 5. Informed Consent as documented by signature or fulfilling the criteria for emergency consent/ deferral consent |
| 6. Agreement of treating physician to perform endovascular procedure |
| **Exclusion criteria** |
| 1. Severe metal artifacts on initial MDCT imaging |
| 2. Planned invasive interventions between MDCT and FDCT scan |
| 3. Clinical deterioration between MDCT and FDCT scan (i.e., an increase of the NIHSS of more than 4 points) |
| 4. Evidence of an ongoing pregnancy prior to randomization. A negative pregnancy test before randomisation is required for all women with child-bearing potential. |

*In case of transfer patients external MDCT images must be available at the PACS of the enrolling center prior to enrollment and the enrolling physician must ensure that the quality of these scans is sufficient for the assessment of the primary endpoint.

FDCT: Flat-Detector CT; MDCT: Multidetector CT; NIHSS: National Institute of Health Stroke Scale.

is present or not) will be determined based on the MDCT scans according to the following procedure: If the assessments of the enrolling physician and of the director of the core-lab (Prof. David Liebeskind, UCLA, Los Angeles, USA) are identical, this will be deemed the ground truth. In cases of disagreement between the site reading and the core lab assessment, the final determination will be made by an expert reader (Prof. Jan Gralla, Inselspital Bern, Bern, Switzerland), who will review unblinded images and available clinical data to reach a final adjudication.

The FDCT images will be read by six readers. Each reader will read all FDCT scans independently. All readings will be done on FDA-approved (compliant with 21CFR892.2050) monitors and according to standard of care settings. A time limit of 90 seconds per FDCT scan will be implemented to simulate real-life conditions. The readers of the MDCT and FDCT scans will not be the same physicians, and they will not share any information. All readings will be documented on standardized forms in a database with an audit trail. The readers were selected by the director of the core lab. All readers must be actively practicing and board certified in one of the following specialties: (neuro-)radiology, vascular neurology or vascular neurosurgery. The Sponsor or Siemens Healthineers AG had no influence on the selection of the readers.

The FDCT readings of each of the six core-lab readers will then be compared with the ground truth. In case at least 5 out of 6 of the first core-lab readers detect a hemorrhage on the FDCT scan, which according to the ground truth was not present on the MDCT scan, Prof. Jan Gralla will use all available images and clinical data (unblinded) to decide if there is any evidence that the hemorrhage occurred between the MDCT and FDCT scans (for example as a complication of intravenous lysis). In this case the patient will be excluded from further analysis (see supplementary table S1 File for detailed information). Based on the results the sensitivity and specificity of non-contrast Sine Spin FDCT imaging for the detection of hemorrhages will be calculated.

## Quality assurance

To ensure compliance with the protocol, and adequate acquisition of the FDCT and MDCT the first five cases at each center will be deemed lead-in cases. They can be excluded and deemed screen failures if they show clear evidence of operator or technical failure based on the assessment of the core lab.

## Sample size calculation

To calculate the sample size, 999 hypothetical experiments were simulated for different combinations of parameters with a sample size ranging from 80 to 260 and with 1–7 readers. We estimate a sensitivity of 99% and a specificity of 98% of non-contrast Sine Spin FDCT imaging for the detection of intracranial hemorrhage as described for the predecessor generation in the literature [7]. Sensitivities and specificities of readers were generated from highly correlated binary random variables to reproduce the correlated readings between readers. To represent patients with an isolated infratentorial intracranial hemorrhage, we introduced a subpopulation of 14 patients (substantial overrepresentation) for whom rater sensitivity was only 70%. In each of the simulated data sets, we assessed whether the lower bound of the 95% CI for sensitivity and specificity remained above the prespecified 95% margins. These CIs were calculated using an intercept-only generalized linear random-effects model with a logit link, incorporating patient and reader as random effects and as outcome the randomly generated binary vector. The "nloptwrap" optimizer ensured model convergence. The experiment was deemed "successful" if both CIs met the margin, and study power was defined as the percentage of successful experiments across 999 resampled datasets. For final sample size determination, we set the number of readers at 6, the sensitivity, specificity, and number of difficult cases as described above, and a 5% drop-out rate. Based on these assumptions and the simulated studies, **252 patients** (238 without drop-outs) must be recruited to achieve a power of 90%, with a 1:1 hemorrhage-positive to negative ratio (126 each, including 14 with isolated infratentorial hemorrhage as part of the hemorrhage-positive group).

## Statistical analysis

To assess the primary endpoint (intracranial hemorrhage), a generalized linear random-effects model with a logit link will be used, with 'reader' and 'patient' as random effects and difficult cases weighted at 2% (i.e., the difficult cases contribute only 2% of the model's effective information, and the normal cases contribute the remaining 98%). A model will be fitted for sensitivity and specificity and model assumptions will be assessed. The 95% CIs for sensitivity and specificity will be derived from the fixed-effect intercept using Wald's method to yield finite, interpretable intervals with expected sensitivity and specificity close to 100%. Sensitivity and specificity will be calculated from core-lab reader assessments of non-contrast Sine Spin FDCT scans, based on hemorrhage-positive and hemorrhage-negative ground truths, respectively. The analysis will test whether the lower bounds of both 95% CIs exceed 95%. The primary analysis will be conducted on the full analysis set (FAS), including all patients without predefined intercurrent events assessed by six readers using a two-point scale (presence or absence of hemorrhage). No missing data are expected, and the excluded patients will be replaced during the study. Fleiss' kappa will measure interrater reliability as a secondary analysis. Specificity, accuracy, positive and negative likelihood ratios will be reported. All analyses will use R (version 4.3.2 or higher) and detailed methods will be described in the statistical analysis plan that will be finalized before the recruitment of the last patient.

## Data safety monitoring committee (DSMC)

An independent DSMC monitors the safety of the trial and meets after 63 participants (25%) and 126 participants (50%) have completed the study to review safety data. All relevant safety variables will be reported. The DSMC has recommended that the trial continues after both meetings.

## Study organization

SPINNERS is an academic investigator-initiated clinical trial. It is sponsored by the University Hospital Basel and the Global Principal Investigator (PI) of the study is Prof. Marios Psychogios. The US PI of the study is Dr. Adam Arthur, MPH. The trial is managed by the Department of Neuroradiology, University Hospital Basel, Switzerland and the Semmes Murphy Foundation, Memphis, USA. The Department of Clinical Research at University Hospital Basel, Switzerland, is

responsible for database oversight, EU site monitoring, and statistical analysis. Monitoring for the US sites is done by the Semmes Murphy Foundation.

### Trial status

Recruitment is ongoing and 229 patients have been enrolled as of July 17, 2025. Completion of enrollment is anticipated in September 2025.

## Discussion

SPINNERS will address the question of whether non-contrast syngo DynaCT Sine Spin FDCT imaging is sufficient to rule out intracranial hemorrhage in suspected acute stroke patients. It will be the first study to examine the diagnostic performance of FDCT for the detection of intracranial hemorrhage prospectively and in the relevant patient population (patients presenting with symptoms suggestive of stroke to the treating hospital).

Previous studies examining the detection of intracranial hemorrhages on FDCT have reported conflicting results [6–8]. A recently published systematic review and meta-analysis suggested a pooled sensitivity of 88% for intraparenchymal hemorrhages and 82% for subarachnoid hemorrhages [9]. A major limitation of these studies is that the patient collective on which the estimates are based is not representative for suspected stroke cohorts. Most studies primarily included patients with peri-interventional hemorrhages such as subarachnoid hemorrhages after EVT, which differ substantially from hemorrhages observed in suspected acute stroke. Therefore, these studies are not suitable to provide an estimate of the diagnostic performance of FDCT.

A distinct strength of the SPINNERS trial lies in its multi-reader design for the blinded assessment of the primary endpoint. While in prior studies the images were read by one or two readers and therefore measuring not only the performance of the method itself but also the performance of the readers, this shortcoming is mitigated in SPINNERS with the use of six independent readers for the FDCT images. As these readers come from different stroke centers in the United States, SPINNERS will give a robust estimate of the diagnostic performance of Sine Spin FDCT imaging across multiple settings. Moreover, this approach increases statistical power while maintaining a feasible sample size.

A potential limitation of the study design is the exclusion of the most severely affected hemorrhagic patients, primarily due to organizational constraints and ethical considerations. These patients often require immediate therapeutic intervention following multidetector computed tomography (MDCT), rendering them ineligible for inclusion in the SPINNERS trial. In addition, minor to moderately affected hemorrhagic patients can be transported more comfortably to the angiography suite for additional imaging as they are more often hemodynamically stable and do not require intubation. This may introduce a selection bias toward less severely affected patients with smaller hemorrhages.

## Conclusion

SPINNERS will inform physicians whether syngo DynaCT Sine Spin FDCT imaging can reliably exclude intracranial hemorrhages in patients with suspected acute stroke.

## Supporting information

**S1 File. SPINNERS_supplementary appendix.** This file includes an overview of sites, investigators and the supplementary tables: **S1 table. Overview of possible outcomes for the readings**; **S2 table. Trial registration: data set**; **S3 table. SPIRIT Checklist for Trials.**
(SPINNERS_SupplementaryAppendix.DOCX)

**S2 File. Protocol version 2.1.** This file includes the final protocol (V2.1 dated 29.04.2024) of the SPINNERS trial.
(DOCX)

**S3 File.  Protocol version 1.2.** This file includes the protocol (V1.2 dated 31.05.2022) at the enrollment of the first patient.
(DOCX)

**S4 File.  Inclusivity-in-global-research-questionnaire.** This file includes the filled-out inclusivity in global research questionnaire.
(DOCX)

**S5. File.  SPIRIT checklist.** This file includes the filled-out SPIRIT Checklist.
(DOCX)

## Author contributions

**Conceptualization:** Marios Psychogios, Alex Brehm, Nitin Goyal, Adam S. Arthur.

**Funding acquisition:** Marios Psychogios, Alex Brehm.

**Investigation:** Marios Psychogios, Alex Brehm, Grégoire Boulouis, Shakeel A. Chowdhry, Donald Frei, Jan Gralla, Johannes Kaesmacher, Ryan T. Kellogg, Christopher P. Kellner, Demetrius K. Lopes, Eytan Raz, Daniel Strbian, Laura Mannismäki, Alejandro Tomasello, Ioannis Tsogkas, Alexander von Hessling, Grzegorz M. Karwacki, Raphael Guzman, Nikki Rommers, David S. Liebeskind, Adam S. Arthur.

**Methodology:** Marios Psychogios, Alex Brehm, Nitin Goyal, Jan-Karl Burkhardt, Adam S. Arthur.

**Project administration:** Marios Psychogios, Alex Brehm, Adam S. Arthur.

**Resources:** Marios Psychogios.

**Supervision:** Alex Brehm.

**Validation:** Alex Brehm.

**Writing – original draft:** Marios Psychogios, Alex Brehm, Adam S. Arthur.

**Writing – review & editing:** Marios Psychogios, Alex Brehm, Nitin Goyal, Grégoire Boulouis, Jan-Karl Burkhardt, Shakeel A. Chowdhry, Donald Frei, Jan Gralla, Johannes Kaesmacher, Ryan T. Kellogg, Christopher P. Kellner, Demetrius K. Lopes, Eytan Raz, Daniel Strbian, Laura Mannismäki, Alejandro Tomasello, Ioannis Tsogkas, Alexander von Hessling, Grzegorz M. Karwacki, Raphael Guzman, Nikki Rommers, David S. Liebeskind.

## Acknowledgments

We thank the SPINNERS investigators: **Marios Psychogios (University Hospital Basel, lead author, marios.psychogios@usb.ch)** Jehuda Soleman (University Hospital Basel), Kristine Ann Blackham (University Hospital Basel), Victor Schulze-Zachau (University Hospital Basel), Anh Nguyen (University Hospital Basel), Nikolaos Ntoulias (University Hospital Basel), Tomas Dobrocky (University Hospital of Bern, Inselspital), Eike Piechowiak (University Hospital of Bern, Inselspital), Sara Pilgram-Pastor (University Hospital of Bern, Inselspital), (University Hospital of Bern, Inselspital),David Seiffge (University Hospital of Bern, Inselspital), Roman Rohner (University Hospital of Bern, Inselspital), Manuel Bolognese (Luzerner Kantonsspital), Lehel Lakatos (Luzerner Kantonsspital), Christian Kamm (Luzerner Kantonsspital), Stephan Bohlhalter (Luzerner Kantonsspital), Marie Guillaume (Luzerner Kantonsspital), Violiza Inoa (Semmes Murphey Clinic and University of Tennessee Health Sciences Center, Memphis), Christopher Nickele (Semmes Murphey Clinic and University of Tennessee Health Sciences Center, Memphis), Nicklaus Khan (Semmes Murphey Clinic and University of Tennessee Health Sciences Center, Memphis), Daniel Hoit (Semmes Murphey Clinic and University of Tennessee Health Sciences Center, Memphis), Lucas Elijovich (Semmes Murphey Clinic and University of Tennessee Health Sciences

Center, Memphis), Visish Srinivasan (Hospital of the University of Pennsylvania, Philadelphia), Josh Catapano (Hospital of the University of Pennsylvania, Philadelphia), Bryan Pukenas (Hospital of the University of Pennsylvania, Philadelphia), Sandeep Kandregula (Hospital of the University of Pennsylvania, Philadelphia), Redi Rahmani (Hospital of the University of Pennsylvania, Philadelphia), Marion Oliver (Advocate Medical Group, Park Ridge), Krishna Joshi (Advocate Medical Group, Park Ridge), Josh Billingsley (Advocate Medical Group, Park Ridge), Kiffon Keigher (Advocate Medical Group, Park Ridge), Joao Victor Sanders (Advocate Medical Group, Park Ridge), William Ares (Endeavor Health, Chicago), Pedro Norat (University of Virginia), Maksim Shapiro (NYU Langone Health, New York), Vera Sharashidze (NYU Langone Health, New York), Erez Nossek (NYU Langone Health, New York), Svetlana Kvint (NYU Langone Health, New York), Rogelio Esparza (NYU Langone Health, New York), Mikko Sillanpää (Helsinki University Hospital), Tatu Kokkonen (Helsinki University Hospital), Riikka Lauha (Helsinki University Hospital), Liisa Tomppo (Helsinki University Hospital), Henrietta Törmänen (Helsinki University Hospital), Manuel Requena (Hospital Vall d'Hebron Barcelona), Francesco Diana (Hospital Vall d'Hebron Barcelona), David Hernandez (Hospital Vall d'Hebron Barcelona),Marta De Dios (Hospital Vall d'Hebron Barcelona), Kevin Janot (CHRU Tours, Tours), Marco Pasi (CHRU Tours, Tours), Fouzi Bala (CHRU Tours, Tours), Nouroudine (Adeniran) Bankole (CHRU Tours, Tours), Abder el Houfia (CHRU Tours, Tours).

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
