## [Decision Letter · Decision Letter 0]

9 Jun 2025

Dear Dr. Brehm,

We look forward to receiving your revised manuscript.

Kind regards,

Stephan Meckel, MD, PhD

Academic Editor

PLOS ONE

Journal Requirements:

“Grant (to University Hospital Basel, MP) from Siemens Healthineers AG for conducting the trial as an investigator initiated trial.

Experts nominated by and employees of Siemens Healthineers AG were consulted during the protocol development, but the Global PI (MP) and the US PI (AA) had the final decision to all suggestions. Siemens Healthineers AG will have no influence on data acquisition, data management, data analyses or scientific interpretation and publication of the results.”

“Grant from Siemens Healthineers AG for conducting the trial as an investigator initiated trial.”

5. One of the noted authors is a group or consortium “SPINNERS investigators”. In addition to naming the author group, please list the individual authors and affiliations within this group in the acknowledgments section of your manuscript. Please also indicate clearly a lead author for this group along with a contact email address.

6. Please include captions for your Supporting Information files at the end of your manuscript, and update any in-text citations to match accordingly. Please see our Supporting Information guidelines for more information: http://journals.plos.org/plosone/s/supporting-information .

7. We note that the original protocol file you uploaded contains a confidentiality notice indicating that the protocol may not be shared publicly or be published. Please note, however, that the PLOS Editorial Policy requires that the original protocol be published alongside your manuscript in the event of acceptance. Please note that should your paper be accepted, all content including the protocol will be published under the Creative Commons Attribution (CC BY) 4.0 license, which means that it will be freely available online, and any third party is permitted to access, download, copy, distribute, and use these materials in any way, even commercially, with proper attribution.

Therefore, we ask that you please seek permission from the study sponsor or body imposing the restriction on sharing this document to publish this protocol under CC BY 4.0 if your work is accepted. We kindly ask that you upload a formal statement signed by an institutional representative clarifying whether you will be able to comply with this policy. Additionally, please upload a clean copy of the protocol with the confidentiality notice (and any copyrighted institutional logos or signatures) removed.

8. We note that the original protocol that you have uploaded as a Supporting Information file contains an institutional logo. As this logo is likely copyrighted, we ask that you please remove it from this file and upload an updated version upon resubmission.

Additional Editor Comments:

**Please address both reviewer's comments in your revision thoroughly, in particular address the uncertainty regarding sample size calculations as well as the statistical model for primary endpoint analysis!**

Reviewers' comments:

Reviewer's Responses to Questions

**Comments to the Author**

1. Does the manuscript provide a valid rationale for the proposed study, with clearly identified and justified research questions?

Reviewer #1: Yes

Reviewer #2: Yes

2. Is the protocol technically sound and planned in a manner that will lead to a meaningful outcome and allow testing the stated hypotheses?

Reviewer #1: Partly

Reviewer #2: Yes

3. Is the methodology feasible and described in sufficient detail to allow the work to be replicable?

Reviewer #1: Yes

Reviewer #2: Yes

4. Have the authors described where all data underlying the findings will be made available when the study is complete?

Reviewer #1: Yes

Reviewer #2: Yes

5. Is the manuscript presented in an intelligible fashion and written in standard English?

Reviewer #1: Yes

Reviewer #2: No

You may also provide optional suggestions and comments to authors that they might find helpful in planning their study.

Reviewer #1: This is an interesting study with a well formulated statistical design and analysis plan with very clear statistical objectives in that the primary Objective is to evaluate if non-contrast syngo DynaCT Sine Spin imaging is non-inferior for the detection of intracranial hemorrhages compared to non-contrast MDCT imaging. It will be deemed non-inferior if the lower bound of the 95%-CI of the sensitivity is above 95% and the lower bound of the 95%-CI of the specificity is above 95% for the detection of intracranial hemorrhages (predefined non-inferiority margin). This is a very routine and clear goal for this endeavor. There are several clarifications needed.

The sample size explanation on page 60 of the protocol is complex It is not quite clear that this simulation of scenario of possible outcomes of number of difficult cases, number of readers at 6, and the drop-out rate at 5% yields a total of 252 patients. Also considering the secondary objective, is it possible that the variation , if any , of the Fleiss kappa impact the study?

What careful planning noted by the authors will be involved to minimize the number of missing data? Specifically detail the planned primary analysis for the FAS. Will it involve assessment of the primary endpoint (intracranial hemorrhage), using only a generalized linear random-effects model with a logit link used, with ‘reader’ and ‘patient’ as random effects and difficult cases weighted at 2%? The 95% CIs for sensitivity and specificity will be derived from the fixed effect intercept using Wald’s method. How will the model be validated as appropriate? Also how will the result be validated (sensitivity and robustness) if an imputed data set is not planned?

The data management, quality assurance and data security are sufficiently detailed and well planned. As an informational query, is there any possibility that the readers may have outcomes that need to be reconciled for a final assessment? If so, what will be the strategy?

Reviewer #2: The paper answers a research question if the proposed approach

However, there are still some comments to the authors that need further refinement.

1. In the introduction part, authors may give the full name of SPINNERS if this is an abbreviated name. Otherwise, the authors may provide a reference.

2. In lines 118-120, the authors should clearly introduce how many buckets are used. And specify the number of participants in each bucket using individual bullets. Authors may consider the term 'groups' instead of 'buckets'

3. Authors should provide a detailed explanation to describe the flow chart for the study design.

4. The author should explain the population before data filtering to provide an overall picture of your study cohort to readers.

5. The authors may provide a clinical characteristic table to demonstrate the study population.

6. The authors might consider a thorough proofread to polish grammar.

**Do you want your identity to be public for this peer review?** For information about this choice, including consent withdrawal, please see our Privacy Policy

Reviewer #1: No

Reviewer #2: No

---

## [Author Response · Author response to Decision Letter 1]

23 Jun 2025

Dear Dr. Meckel,

First, we want to express our gratitude for your and the reviewer’s evaluation of the manuscript. Important points were raised, which improve the quality of the manuscript.

We have updated the trial status on page 10 line 261 – 262 to reflect the status.

Please find attached a point-by-point reply to the points raised:

Journal Requirements:

Thank you for raising this important point, we have adapted the manuscript and file naming accordingly.

Thank you very much to pointing us to this important questionnaire. We have filled out the questionnaire and uploaded it with the revision.

3. Thank you for stating the following financial disclosure: “Grant (to University Hospital Basel, MP) from Siemens Healthineers AG for conducting the trial as an investigator-initiated trial. Experts nominated by and employees of Siemens Healthineers AG were consulted during the protocol development, but the Global PI (MP) and the US PI (AA) had the final decision to all suggestions. Siemens Healthineers AG will have no influence on data acquisition, data management, data analyses or scientific interpretation and publication of the results.”

Thank you for bringing this up. In our opinion the financial disclosure statement correctly reflects the role of the funder in the trial. Employees of the funder (Siemens Healthineers AG) and experts nominated by the funder were consulted during the protocol development stage, however all decision were made by the Global PI (MP) and the US PI (AA). However, for clarity we would slightly reformulate the statement:

“A grant from Siemens Healthineers AG was given to the University Hospital Basel (Sponsor Investigator Marios Psychogios) for conducting the trial as an investigator-initiated trial. Durin the planning phase of the trial experts nominated by and employees of Siemens Healthineers AG were consulted, but the Global PI (Marios Psychogios) and the US PI (Adam Arthur) had the final decision to all suggestions. Siemens Healthineers AG had no role in the collection, monitoring, analysis, and interpretation of the data or scientific interpretation and publication of the results. This does not alter our adherence to PLOS ONE policies on sharing data and materials.”

We sincerely hope this addresses the concerns raised.

4. Thank you for stating the following in the Competing Interests section: “Grant from Siemens Healthineers AG for conducting the trial as an investigator initiated trial.” Please confirm that this does not alter your adherence to all PLOS ONE policies on sharing data and materials, by including the following statement: "This does not alter our adherence to PLOS ONE policies on sharing data and materials.” (as detailed online in our guide for authors http://journals.plos.org/plosone/s/competing-interests). If there are restrictions on sharing of data and/or materials, please state these. Please note that we cannot proceed with consideration of your article until this information has been declared.

Thank you for bringing this up. We confirm that this does not alter our adherence to PLOS ONE policies on sharing data and materials. We have included this in the updated statement (see answer under point 3).

5. One of the noted authors is a group or consortium “SPINNERS investigators”. In addition to naming the author group, please list the individual authors and affiliations within this group in the acknowledgments section of your manuscript. Please also indicate clearly a lead author for this group along with a contact email address.

We have added this under acknowledgments. Thank you for bringing this up.

We have included the captions and updated them accordingly.

7. We note that the original protocol file you uploaded contains a confidentiality notice indicating that the protocol may not be shared publicly or be published. Please note, however, that the PLOS Editorial Policy requires that the original protocol be published alongside your manuscript in the event of acceptance. Please note that should your paper be accepted, all content including the protocol will be published under the Creative Commons Attribution (CC BY) 4.0 license, which means that it will be freely available online, and any third party is permitted to access, download, copy, distribute, and use these materials in any way, even commercially, with proper attribution. Therefore, we ask that you please seek permission from the study sponsor or body imposing the restriction on sharing this document to publish this protocol under CC BY 4.0 if your work is accepted. We kindly ask that you upload a formal statement signed by an institutional representative clarifying whether you will be able to comply with this policy. Additionally, please upload a clean copy of the protocol with the confidentiality notice (and any copyrighted institutional logos or signatures) removed.

We have uploaded a clean version of the protocol with the confidentiality note crossed out and a letter confirming publication under CC BY 4.0 is permitted given the manuscript is accepted for publication.

8. We note that the original protocol that you have uploaded as a Supporting Information file contains an institutional logo. As this logo is likely copyrighted, we ask that you please remove it from this file and upload an updated version upon resubmission.

We have removed the logo in the new version.

Reviewer #1:

Reviewer #1: This is an interesting study with a well formulated statistical design and analysis plan with very clear statistical objectives in that the primary Objective is to evaluate if non-contrast syngo DynaCT Sine Spin imaging is non-inferior for the detection of intracranial hemorrhages compared to non-contrast MDCT imaging. It will be deemed non-inferior if the lower bound of the 95%-CI of the sensitivity is above 95% and the lower bound of the 95%-CI of the specificity is above 95% for the detection of intracranial hemorrhages (predefined non-inferiority margin). This is a very routine and clear goal for this endeavor. There are several clarifications needed.

Thank you for kind summary of the study, please see below answers to the points raised.

The sample size explanation on page 60 of the protocol is complex It is not quite clear that this simulation of scenario of possible outcomes of number of difficult cases, number of readers at 6, and the drop-out rate at 5% yields a total of 252 patients. Also considering the secondary objective, is it possible that the variation , if any , of the Fleiss kappa impact the study?

We apologize for the unclarity to the reviewer. Indeed, the simulation scenario with 6 independent readers, 14 difficult cases (with a sensitivity of 70%) and an assumed sensitivity of 99% and specificity of 98% yielded a total sample size of 252 patients including drop out based on 999 simulated experiments. We have now clarified those assumptions in the respective section of the paper. The distribution of the number of positive and negative hemorrhagic patients, as well as the number of difficult cases are predefined, and patients are recruited according to the bins. Therefore, the variability of sample sizes is fixed.

In the simulations, we assumed a moderate positive correlation among readers to reflect similar experience levels. This introduces a baseline level of agreement above chance, which does elevate Fleiss’ kappa values somewhat. However, it is important to note that Fleiss’ kappa is not determined by correlation alone. It is primarily influenced by the overall pattern of agreement across readers and the distribution of category ratings. Given the expected high performance (i.e., near-perfect sensitivity and specificity), disagreement is mostly concentrated in the small subset of “difficult” cases. Consistent ratings in these cases have a disproportionately large influence on the kappa statistic. Our setup allows us to estimate Fleiss’ kappa under plausible clinical conditions without biasing the results through over- or underestimation of inter-reader agreement.

Furthermore, the assumed correlation also impacts the statistical power of the study. A very high correlation reduces the effective independence of the readers, lowering variability and thereby decreasing the power to detect differences in agreement. Conversely, assuming no correlation would overestimate power and risk underpowering the study in practice. We selected six readers to strike a balance between sufficient replication per patient (to enable robust estimation of sensitivity, specificity, and kappa) and avoiding excessive redundancy due to correlation. Our simulations confirmed that under the assumed parameters, this design yields adequate power to meet our primary study objective.

What careful planning noted by the authors will be involved to minimize the number of missing data? Specifically detail the planned primary analysis for the FAS. Will it involve assessment of the primary endpoint (intracranial hemorrhage), using only a generalized linear random-effects model with a logit link used, with ‘reader’ and ‘patient’ as random effects and difficult cases weighted at 2%? The 95% CIs for sensitivity and specificity will be derived from the fixed effect intercept using Wald’s method. How will the model be validated as appropriate? Also how will the result be validated (sensitivity and robustness) if an imputed data set is not planned?

We thank the reviewer for raising these points regarding the primary analysis.

Minimizing the number of missing data:

This study takes place in an emergency setting, and it includes two scans that should be taken within 4 hours of one another. There is no follow-up visit and no foreseeable scenario in which a patient might be lost to follow-up. All scans are rated by an independent core-lab after recruitment has been completed and the primary analysis only includes the readings of the scans. Therefore, no missing data are expected.

There are two intercurrent events defined which would lead to missing data:

• Indeterminate scans: this is avoided as much as possible through thorough training of the participating study centers, and a lead-in phase consisting of the first five cases from each site. However, indeterminate scans may still occur due to operating failures (e.g., clear artefacts due to objects in the field of view, wrong positioning of the head) or technical failure due to missing calibration of the scanner. In accordance with the protocol, those cases are identified by the independent core-lab and those patients are excluded from analysis. Excluded cases are replaced to ensure adequate power.

• Intracranial hemorrhage between the two scans: this is a rare, but possible intercurrent event. The consequence of this intercurrent event is a correct reading of no hemorrhage as the ground truth, and a hemorrhage identified on the FDCT scans which is also correct. The possibility of a hemorrhage between the two scans is investigated in case of a discrepancy in the ground truth and at least 5 out of 6 readers of the independent core-lab. In this case, a second independent core-lab evaluates all images and if there is a possibility of an intracranial hemorrhage that occurred between the two scans, the patient is excluded from analysis. This approach avoids biasing of false positives.

Since the scans are rated by an independent core-lab and given the exclusion of patients with intercurrent events, we do not expect missing data in the primary analysis. All analysis is based on FAS with all patients without intercurrent events as described above.

Primary analysis

We estimate sensitivity and specificity of non-contrast FDCT for the detection of intracranial hemorrhage using generalized mixed‑effects model (GLMM) with a logit link function and binomial distribution. The model is fit once in patients with and once in patients without hemorrhage, as defined by the ground truth. The model’s estimated marginal probability in patients with hemorrhage represents sensitivity, while the estimated marginal probability in patients without hemorrhage represents specificity. Models are fit with the glmer() function from the R package lme4.

We use the “nloptwrap” optimizer to improve convergence in near‑separation situations that we expect due to the assumed very high sensitivity and specificity. Difficult cases are weighted with 2% (i.e., the difficult cases contribute only 2% of the model’s effective information, and the normal cases contribute the remaining 98%). Specific weights are calculated by dividing the mass of difficult cases (n*0.02) by the total number of difficult cases. In the same way, we calculate the weight for the normal cases, using 98% as a contribution.

From the fixed effect intercept of sensitivity and specificity, the 95% confidence interval (CI) is calculated applying the method of Wald. Although profile‑likelihood or bootstrap CIs can be used, near‑boundary estimates (sensitivities/specificities close to 100%) can lead to non‑finite profile intervals or numerical instabilities. The Wald CI on the log‑odds scale is guaranteed to yield finite, interpretable intervals on the probability scale.

We test whether the lower bounds of the 95% CI of both the sensitivity and the specificity are both higher than the pre-specified margin of 95%. Non‑inferiority (i.e. performance above the threshold) is declared if the lower bound of the two‑sided 95% CI for both sensitivity and specificity exceeds 0.95. This corresponds to a one‑sided alpha = 0.025. Since sensitivity and specificity are treated as co-primary endpoints, testing both at the same alpha level does not reduce power or require correction.

Validation of the model of the primary analysis

To ensure the model is appropriate and well-specified, we will undertake the following steps:

• Model assumption check: we will verify model convergence and examine fit diagnostics such as residuals and random effect distribution, and influential observations. We will also inspect the estimated variances for the random effect to ensure meaningful contributions and no overfitting.

• Sensitivity analysis: We do not expect missing data for the primary analysis, but we might exclude patients. We will compare the baseline characteristics of the excluded patients with the included patients to understand the potential bias. We will also assess the robustness of the non-inferiority conclusions with other link funct

---

## [Decision Letter · Decision Letter 1]

9 Jul 2025

Dear Dr. Brehm,

Thank you for submitting your manuscript to PLOS ONE. After careful consideration, we feel that it has merit but does not fully meet PLOS ONE’s publication criteria as it currently stands. Therefore, we invite you to submit a revised version of the manuscript that addresses the points raised during the review process.

We look forward to receiving your revised manuscript.

Kind regards,

Stephan Meckel, MD, PhD

Academic Editor

PLOS ONE

Journal Requirements:

Additional Editor Comments:

**Please address reviewer 2's additional comment.**

Reviewers' comments:

Reviewer's Responses to Questions

**Comments to the Author**

1. Does the manuscript provide a valid rationale for the proposed study, with clearly identified and justified research questions?

Reviewer #1: Yes

Reviewer #2: Yes

2. Is the protocol technically sound and planned in a manner that will lead to a meaningful outcome and allow testing the stated hypotheses?

Reviewer #1: Yes

Reviewer #2: Yes

3. Is the methodology feasible and described in sufficient detail to allow the work to be replicable?

Reviewer #1: Yes

Reviewer #2: Yes

4. Have the authors described where all data underlying the findings will be made available when the study is complete?

Reviewer #1: Yes

Reviewer #2: Yes

5. Is the manuscript presented in an intelligible fashion and written in standard English?

Reviewer #1: Yes

Reviewer #2: Yes

You may also provide optional suggestions and comments to authors that they might find helpful in planning their study.

Reviewer #1: The queries have been addressed adequately. Please check for typos. For example , on the revision, page 10, line 251 'Durin' should be 'During'.

Reviewer #2: Thanks for addressing the previous comments and questions. I also have some comments for the current version:

1. In ischemic and hemorrhagic stroke, the authors may provide the definition such as ICD-10 or some other source for disease diagnosis.

**Do you want your identity to be public for this peer review?** For information about this choice, including consent withdrawal, please see our Privacy Policy

Reviewer #1: No

Reviewer #2: No

---

## [Author Response · Author response to Decision Letter 2]

21 Jul 2025

Dear Dr. Meckel,

First, we want to express our gratitude for your and the reviewer’s evaluation of the manuscript. Important points were raised, which improve the quality of the manuscript.

We have updated the trial status on page 10 line 261 – 262 to reflect the status.

Please find attached a point-by-point reply to the points raised:

Journal Requirements:

We have reviewed the reference list and ensured its completeness. One reference was incorrect numbered; we have corrected this. In specific in this sentence reference 6 was included but 7 was the correct reference:

“We estimate a sensitivity of 99% and a specificity of 98% of non-contrast Sine Spin FDCT imaging for the detection of intracranial hemorrhage as described for the predecessor generation in the literature. [7]”

Reviewer #1:

The queries have been addressed adequately. Please check for typos. For example , on the revision, page 10, line 251 'Durin' should be 'During'.

Thank you very much for highlighting this typo. We worked with a native speaker through the manuscript to improve grammar and correct all typos.

Reviewer #2:

Thanks for addressing the previous comments and questions. I also have some comments for the current version:

1. In ischemic and hemorrhagic stroke, the authors may provide the definition such as ICD-10 or some other source for disease diagnosis.

Thank you for your suggestion. In our study, the diagnosis of ischemic and hemorrhagic stroke is made in the acute setting, primarily in the emergency room. At this stage, clinical evaluation (such as an acute neurological deficit assessed using NIHSS or GCS) combined with imaging findings on multidetector CT (MDCT) forms the basis for diagnosis. Since these assessments occur before formal coding and discharge documentation, referencing an ICD-10 code is not appropriate in this context.

Furthermore, our inclusion criteria focus on patients who present via a pre-hospital stroke alert and are received as suspected acute stroke cases. This approach ensures timely triage and intervention, which is critical in the acute management of stroke.

We have also clarified this in the protocol paper and added under “Population” (Page 6 line 149 – 153) the following sentence: “The diagnosis of an ischemic or hemorrhagic stroke is made in the acute setting, primarily in the emergency room, prior to patient enrollment. At this stage, clinical evaluation—such as the assessment of acute NIHSS or the Glasgow Coma Scale (GCS) — combined with imaging findings from the MDCT forms the basis for diagnosis.”

Thank you for considering the manuscript for publication,

Marios Psychogios and Adam S Arthur

---

## [Decision Letter · Decision Letter 2]

5 Aug 2025

ProSPective evaluation of the dIagnostic accuracy of siNe spiN non-contrast flat-dEtectoR CT (FDCT) for the detection of intracranial hemorrhage in stroke patients - Protocol of a non-inferiority comparison to multi detector CT

PONE-D-25-10196R2

Dear Dr. Brehm,

We’re pleased to inform you that your manuscript has been judged scientifically suitable for publication and will be formally accepted for publication once it meets all outstanding technical requirements.

Kind regards,

Stephan Meckel, MD, PhD

Academic Editor

PLOS ONE

Additional Editor Comments (optional):

Reviewers' comments:

Reviewer's Responses to Questions

**Comments to the Author**

1. Does the manuscript provide a valid rationale for the proposed study, with clearly identified and justified research questions?

Reviewer #1: Yes

2. Is the protocol technically sound and planned in a manner that will lead to a meaningful outcome and allow testing the stated hypotheses?

Reviewer #1: Yes

3. Is the methodology feasible and described in sufficient detail to allow the work to be replicable?

Reviewer #1: No

4. Have the authors described where all data underlying the findings will be made available when the study is complete?

Reviewer #1: Yes

5. Is the manuscript presented in an intelligible fashion and written in standard English?

Reviewer #1: Yes

You may also provide optional suggestions and comments to authors that they might find helpful in planning their study.

Reviewer #1: XXXXXXXXXXXXXXXXXXXXXXXXXXXXXXXXXXXXXXXXXXXXXXXXXXXXXXXXXXXXXXXXXXXXXXXXXXXXXXXXXXXXXXXXXXXXXXXXXXXXXXXXXXXXXXXXXXXXXXXXXXXXXXXXX

**Do you want your identity to be public for this peer review?** For information about this choice, including consent withdrawal, please see our Privacy Policy

Reviewer #1: No

---

## [Editor Report · Acceptance letter]

PONE-D-25-10196R2

PLOS ONE

Dear Dr. Brehm,

I'm pleased to inform you that your manuscript has been deemed suitable for publication in PLOS ONE. Congratulations! Your manuscript is now being handed over to our production team.

Kind regards,

on behalf of

Prof. Dr. Stephan Meckel

Academic Editor

PLOS ONE